



# High-performance coupled surface-subsurface flow simulation with SERGHEI-SWE-RE

Na Zheng[1], Zhi Li[1], Gregor Rickert[2], Mario Morales-Hernández[3], Ilhan Özgen-Xian[2,4], and
Daniel Caviedes-Voullième[5,6]

[1]College of Civil Engineering, Tongji University, Shanghai, China
[2]Institute of Geoecology, Technische Universität Braunschweig, Brunswick, Germany
[3]Fluid Mechanics, I3A, Universidad de Zaragoza, Zaragoza, Spain
[4]Leichtweiß-Institute for Hydraulic Engineering and Water Resources, Technische Universität Braunschweig, Brunswick, Germany
[5]Simulation and Data Lab Terrestrial Systems, Jülich Supercomputing Centre, Forschungszentrum Jülich, Jülich, Germany
[6]Institute of Bio- and Geosciences Agrosphere (IBG-3), Forschungszentrum Jülich, Jülich, Germany

**Correspondence:** Zhi Li (zli90@tongji.edu.cn)

**Abstract.** This work presents SERGHEI-SWE-RE, a performance-portable, parallel model that couples a fully dynamic two-dimensional Shallow Water Equation (SWE) solver with a three-dimensional Richards Equation (RE) solver within the Kokkos framework to simulate surface–subsurface flow exchange. The model features a modular architecture with sequential coupling strategy, supporting both synchronous and asynchronous executions of surface and subsurface modules. The SERGHEI-SWE-RE model is validated against five benchmark problems incorporating stationary and fluctuating free-surface tests, a tilted v-catchment, a lateral-flow slope without ponding, and a heterogeneous superslab. The results demonstrate good agreement with established models. Asynchronous coupling reduces wall-clock time by up to about 60% in the superslab case while preserving simulation accuracy. Strong and weak scaling tests on multiple Intel Xeon CPUs and NVIDIA GPUs reveal robust portability, with near-ideal RE scaling and less-satisfactory SWE scaling at high GPU counts, suggesting future improvements on differentiated meshes or more advanced domain decomposition strategies. Overall, the results presented establish SERGHEI-SWE-RE as an efficient, flexible and scalable model for integrated surface-subsurface flow simulations.

## 1 Introduction

Surface-subsurface water exchange (SSE) is a critical process in the hydrological cycle, which exerts a substantial influence on the dynamics of the water budget and water quality (Dennedy-Frank, 2019; Ntona et al., 2022). In particular, SSE determines in which compartment water volumes are, which in turn has a dominant effect on the time scales of the dynamics of such water volumes. Integrated hydrological models (IH models) have been widely used to simulate SSE across diverse scenarios (Paniconi and Putti, 2015; Fatichi et al., 2016), including river basins (e.g., Aliyari et al., 2019; Nixdorf et al., 2017; Gilbert and Maxwell, 2017), lake watersheds (e.g., Autio et al., 2023; Ala-aho et al., 2015; Persaud et al., 2020) and coastal regions (e.g., Daneshmand et al., 2019; Guimond and Michael, 2021; Li et al., 2023; Paldor et al., 2022; Xiao et al., 2017; Yang et al., 2013; Zhang et al., 2018). Valuable insights are also gained from studies on the surface-subsurface coupling algorithms (e.g., Chen



et al., 2022; Liggett et al., 2013; de Rooij, 2017). A comprehensive review and comparison of existing IH models can be found in Shu et al. (2024). Moreover, IH models are increasingly being integrated with geochemical, ecological and land surface models, facilitating scientific research across a broad range of topics in earth and environmental sciences (e.g., Forrester and Maxwell, 2020; Hein et al., 2019; Naz et al., 2023; Zhang et al., 2024).

In IH models, the Shallow Water Equation (SWE)—or its variants—and the Richards equation (RE) (Richards, 1931) are typically employed as the governing equations for surface and subsurface flow, respectively (Haque et al., 2021; Shu et al., 2024). Depending on the coupling strategy, these models can be broadly classified into two categories: fully coupled and sequentially coupled models. Fully coupled models, such as Parflow (Kollet and Maxwell, 2006), HGS (Brunner and Simmons, 2012; Tang et al., 2024), WASH123D (Yeh et al., 1998; Hussain, 2022), ATS (Coon et al., 2020; Jan et al., 2020), solve
the surface and subsurface flow equations simultaneously in a single system (Morita and Yen, 2002; Huang and Yeh, 2009; Shu et al., 2024). However, this coupling strategy could be computationally inefficient due to the mismatch between surface and subsurface time scales. That is, surface flow processes are typically faster and more dynamic than subsurface flow. For example, the speed of supercritical shallow water flow could exceed 1 m/s, but the hydraulic conductivity of the subsurface soil is usually less than 0.001 m/s (Furnari et al., 2024). This suggests that the subsurface flow solver could potentially use
a larger numerical time step ($\Delta t$) than the surface flow solver, without compromising model stability. The fully coupled IH model requires a unified $\Delta t$ for both surface and subsurface flow, thereby limiting the overall computational speed. In contrast, sequentially coupled models solve surface and subsurface flow equations separately, and the SSE is explicitly computed during synchronization to ensure coherence and continuity in the overall simulation. Notable examples include CATHY (Camporese et al., 2010), OpenGeoSys (Delfs et al., 2012), inHM (VanderKwaak and Loague, 2001), and tRIBS+OFM (Kim et al., 2012).
This approach offers greater flexibility when modeling either surface or subsurface flow independently (Huang and Yeh, 2009; Shu et al., 2024). Moreover, it allows different $\Delta t$ for surface and subsurface calculations, potentially improving computational efficiency. For example, Li et al. (2023) demonstrated that asynchronous coupling can improve computational efficiency by up to 81% compared to fully coupled approaches.

    IH models are typically computationally demanding because of the scale of the study area, especially when the fully dynamic
two-dimensional (2D) SWE and the three-dimensional (3D) RE are used. As an alternative, simplified SWEs, such as the diffusive wave or the kinematic wave equation, are widely used to describe surface flow (Maxwell et al., 2014). While these simplifications enable larger $\Delta t$, they have limited applicability for rapidly varying flows, such as floods or tidal oscillations (Li and Hodges, 2021). Similarly, some models assume that the subsurface flow is 1D in the unsaturated zone to improve computational efficiency (Graham and Refsgaard, 2001; Gunduz and Aral, 2005; Kong et al., 2010), but this approximation
is inadequate to address pronounced lateral flow in the vadose zone (Mao et al., 2021). Therefore, solving the fully dynamic 2D SWE and 3D RE is indispensable for comprehensive IH models capable of accurately simulating surface hydrodynamics, variable-saturated subsurface flow and surface-subsurface flow exchange under various application scenarios.

    Recent advances in High-Performance Computing (HPC) technology provide a viable pathway for integrating fully dynamic 2D SWE solvers and 3D RE solvers with acceptable computational cost (Bhanja et al., 2023). Several HPC-based models have
been developed and validated through idealized numerical tests and watershed-scale applications (Le et al., 2015; Kuffour et al.,





2020; Wu et al., 2021). However, the diversity of modern supercomputing architectures poses challenges for cross-platform portability and broader adoption of these models (Hokkanen et al., 2021). To address these issues, heterogeneous programming models have emerged as effective solutions, offering code portability and hardware independence (Fang et al., 2020). Among these, the Kokkos framework enables performance portability across various manycore architectures for models by providing a unified abstraction for data parallelism and memory access patterns (Edwards et al., 2014). Leveraging Kokkos, Caviedes-Voullieme et al. (2023) developed the SERGHEI modeling framework. SERGHEI is an an open-source, high-performance, and performance-portable platform under active development for simulating integrated hydrodynamic, ecohydrologic, and geomorphologic processes. Key components of SERGHEI include a fully dynamic 2D Shallow Water Equation (SWE) solver (SERGHEI-SWE) (Caviedes-Voullieme et al., 2023) and a 3D Richards Equation (RE) solver (SERGHEI-RE)(Li et al., 2025). However, the capability of simulating SSE using SERGHEI has yet to be explored and demonstrated.

In this study, we introduce the integrated surface-subsurface flow simulation capabilities of SERGHEI and demonstrate its simulation accuracy, as well as parallel scalability. The integrated model, referred as SERGHEI-SWE-RE, offers several key advantages:

1. Full dimension, dynamic flow processes. SERGHEI-SWE-RE combines the fully dynamic 2D SWE and 3D RE to simulate coupled surface and subsurface flow. The adoption of these complete governing equations ensures the applicability of SERGHEI-SWE-RE under a wide range of scenarios with various flow conditions.

2. Modularization and flexibility. SERGHEI-SWE-RE uses a sequential surface-subsurface coupling approach and allows asynchronous coupling (see Section 2.2 for further details). It offers the flexibility to simulate the integrated system, as well as the surface or the subsurface flow alone. Incorporating asynchronous coupling further enhances the computational efficiency of the integrated simulation.

3. Computational efficiency and portability. SERGHEI-SWE-RE supports performance-portable parallel computation on a variety of computational platforms through the Kokkos framework.

The paper is structured as follows: Section 2 describes the governing equations used in SERGHEI-SWE-RE and explains emphatically the details of the coupling strategy. Section 3 and 4 present the numerical cases and a real-world scenario to support the performance of the integration model. Section 5 presents the conclusions of this paper.

## 2 Numerical methods

### 2.1 Surface and subsurface flow

The numerical methods used in SWE and RE modules of SERGHEI have been described in detail in Caviedes-Voullieme et al. (2023) and Li et al. (2025), respectively. Herein, we provide a concise overview of the governing equations, numerical schemes, and key solution features.





In SERGHEI-SWE, surface flow is modeled by solving the fully dynamic 2D SWE using a finite volume method for spatial discretization combined with an explicit Euler method for temporal integration. In vector form, the 2D SWE reads

$$
\begin{cases}
\dfrac{\partial \boldsymbol{U}}{\partial t} + \dfrac{\partial \boldsymbol{F}}{\partial x} + \dfrac{\partial \boldsymbol{G}}{\partial y} = \boldsymbol{S}_r + \boldsymbol{S}_b + \boldsymbol{S}_f, \\[2mm]
\boldsymbol{U} = \begin{bmatrix} h \\ q_x \\ q_y \end{bmatrix}, \boldsymbol{F} = \begin{bmatrix} q_x \\ \frac{q_x^2}{h} + \frac{1}{2}gh^2 \\ \frac{q_x q_y}{h} \end{bmatrix}, \boldsymbol{G} = \begin{bmatrix} q_y \\ \frac{q_x q_y}{h} \\ \frac{q_y^2}{h} + \frac{1}{2}gh^2 \end{bmatrix}, \\[2mm]
\boldsymbol{S}_r = \begin{bmatrix} r_o - r_f \\ 0 \\ 0 \end{bmatrix}, \boldsymbol{S}_b = \begin{bmatrix} 0 \\ -gh\frac{\partial z}{\partial x} \\ -gh\frac{\partial z}{\partial y} \end{bmatrix}, \boldsymbol{S}_f = \begin{bmatrix} 0 \\ -gh\sigma_x \\ -gh\sigma_y \end{bmatrix}.
\end{cases}
\tag{1}
$$

where $t$ denotes time $[T]$; $x, y$ represent the Cartesian coordinates $[L]$; $h$ signifies the pressure head $[L]$; The components $q_x = hu$ and $q_y = hv$ represent the unit discharges in $x$ and $y$ directions $[L^2 T^{-1}]$, respectively; $g$ is the gravitational acceleration $[LT^{-2}]$; $r_o$ represents the rainfall intensity $[LT^{-1}]$ and $r_f$ is the infiltration (or exfiltration) rate $[LT^{-1}]$; $\frac{\partial z}{\partial x}, \frac{\partial z}{\partial y}$ are the bed slope in $x$ and $y$ directions; $\sigma_x, \sigma_y$ are the friction slope in $x$ and $y$ directions. The shallow water equation (Eq. 1) is solved with a first-order finite volume scheme (Morales-Hernández et al., 2021). The stability of the scheme is constrained by the Courant–Friedrichs–Lewy (CFL) condition, where a CFL number less than 0.5 is required.

In SERGHEI-RE, subsurface flow is described using the generalized 3D RE as

$$
\frac{\partial \theta}{\partial t} + \frac{S_s \theta}{\phi} \frac{\partial h}{\partial t} - \nabla \cdot (\boldsymbol{K}(h)\nabla(h - z)) - q_s = 0
\tag{2}
$$

where $\theta$ is the volumetric water content; $S_s$ is the specific storage capacity $[L^{-1}]$; $\phi$ is porosity; $h$ is pressure head $[L]$; $\boldsymbol{K}$ indicates the hydraulic conductivity tensor $[LT^{-1}]$, which is a function of the pressure head; $z$ represents the elevation head $[L]$; $q_s$ incorporates source/sink terms $[T^{-1}]$. It should be noted that this formulation includes the specific storage effects to unify the modeling of both saturated and unsaturated subsurface flow regimes (Krabbenhøft, 2007).

Equation (2) requires a model closure, which is achieved by means of soil-water retention models. In SERGHEI-RE, the popular Mualem-van Genuchten model (Mualem, 1976; van Genuchten, 1980) is adopted to describe the relationship between the pressure head ($h$), the water content ($\theta$), and the hydraulic conductivity ($K$) as

$$
\begin{cases}
\theta(h) = \theta_r + \dfrac{(\theta_s - \theta_r)}{(1 + |\alpha h|^n)^m}, \\[2mm]
K(h) = K_s (1 + |\alpha h|^n)^{-\frac{m}{2}}(1 - (1 - (1 + |\alpha h|^n)^{-1})^m)^2, \\[2mm]
m = 1 - \dfrac{1}{n},
\end{cases}
\tag{3}
$$

where the model parameters include the residual water content ($\theta_r$), the saturated water content ($\theta_s$), the saturated hydraulic conductivity ($K_s$ $[LT^{-1}]$), and two soil-specific parameters ($\alpha$ $[L^{-1}]$ and $n$).

In SERGHEI-RE, the RE is spatially discretized by a finite volume scheme. For time integration, two numerical schemes are available: (i) the predictor-corrector scheme that shows superior convergence behavior, but requires a smaller time step





size ($\Delta t$), and (ii) the modified Picard iterative scheme that allows for large time steps, but may fail to converge under certain hydrogeological conditions. For a detailed comparison of the two schemes, refer to Li et al. (2024).

## 2.2 Surface-subsurface exchange

In SERGHEI-SWE-RE, the surface and subsurface domains share the same horizontal discretization. When SSE occurs, each surface cell communicates with one subsurface cell through a *halo* layer (Fig. 1). After solving the SWE, the surface water depth is transferred to the *halo* layer, serving as either pressure head or flux boundary conditions (BC) for the subsurface domain. Subsequently, after solving the RE, the updated SSE flux is converted to an equivalent water depth and added directly to the surface water depth. The complete SSE computation procedure is shown in Fig. 2 and described as follows:

1. Solve the SWE (excluding SSE) to obtain the surface water depth ($h_s^*$), and transfer it to the *halo* layer. Note that "$*$" indicates an intermediate solution of the water depth.

2. Determine the type of boundary condition to be applied to each cell in the top layer of the subsurface domain. To complete this step, SERGHEI first calculates a hypothetical SSE flux ($q_{ss}^*$) based on Darcy's Law (Eq. 4) as

$$q_{ss}^* = 2 \times K \frac{h_g - h_s^*}{\Delta z} - K, \tag{4}$$

where $\Delta z$ is the height of the top subsurface cell and upward flux is marked positive. Then, there are four possible situations:

   (i) If the surface cell is wet, and $-q_{ss}^* \times \Delta t <= h_s^*$, indicating that either exfiltration occurs, or the infiltration volume is less that the available surface water, the *halo* layer ($h_s^*$) is used as the pressure head boundary condition for the subsurface domain (Fig. 2b).

   (ii) If the surface cell is wet, but $-q_{ss}^* \times \Delta t > h_s^*$, indicating that all surface water infiltrate into the subsurface within the current time step. In this case, using a pressure head boundary condition would overestimate infiltration. Instead, a flux boundary condition is applied to the top subsurface layer, where the SSE flux is set to the equivalent flux of the available ponding depth. (Fig. 2c).

   (iii) If the surface cell is dry, and $h_g - \Delta z/2 > h_s^*$, indicating that exfiltration occurs, similar to condition (i), the *halo* layer ($h_s^*$) is used as the pressure head boundary condition (Fig. 2d).

   (iv) If the surface cell is dry, but $h_g - \Delta z/2 <= h_s^*$, indicating that neither infiltration nor exfiltration should occur, a no flow boundary condition is enforced to the subsurface domain (Fig. 2e).

3. Once the type of subsurface boundary condition is determined, solve the RE to update the pressure head.

4. Calculate the final SSE flux, $q_{ss}$. For Case (i) and (iii), $q_{ss}$ is determined using Darcy's Law, similar to Eq. (4). For Case (ii), $q_{ss} = -h_s^*/\Delta t$. For Case (iv), $q_{ss} = 0$.





5. Calculate an equivalent SSE depth from the SSE flux, and add it to the surface water depth to get the final surface water depth. That is, $h_s = h_s^* + \Delta t \times q_{ss}$.

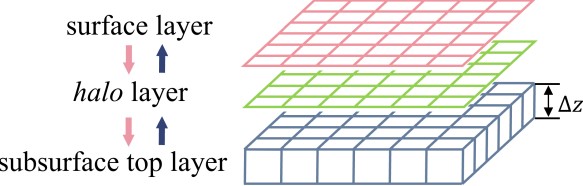

**Figure 1.** The schematic of data exchange between subdomains.

A special case is the rainfall condition. When the ground surface is dry, rainfall is typically a flux boundary condition to the subsurface domain. In SERGHEI-SWE-RE, for simplicity, it is assumed that rainfall is always applied to the surface domain, accumulates as surface ponding, then infiltration is computed following the above procedure. This avoids the need to handle rainfall in the subsurface solver. The validity of this approach will be demonstrated in Section 3.4.

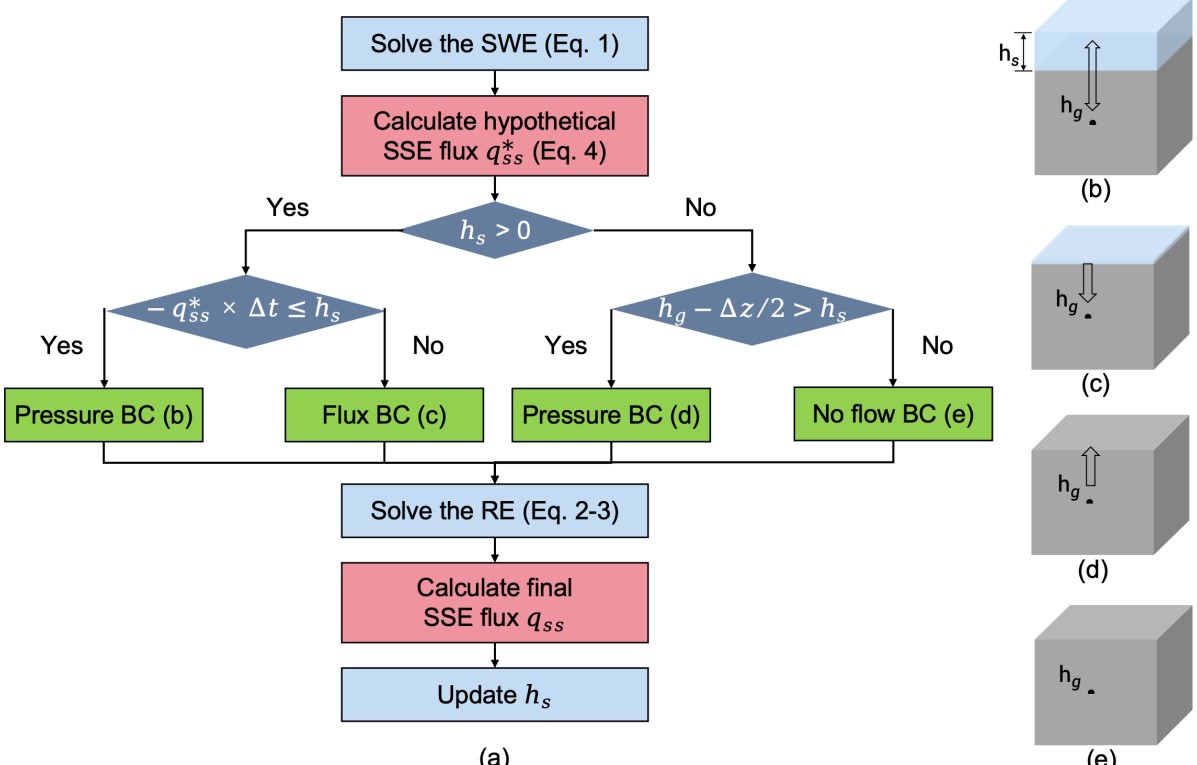

**Figure 2.** (a) The flow chart of the SSE simulation process. (b)-(e) Illustrations of the four possible situations where the SSE fluxes are calculated differently.



SERGHEI-SWE-RE employs a modular structure to maximize model flexibility across a wide range of application scenarios.
It supports asynchronous coupling between surface and subsurface flow simulations. As illustrated in Fig. 3, the surface and subsurface modules operate on their independent timelines ($t_{\text{sur}}$ and $t_{\text{sub}}$) with their own time step lengths ($\Delta t_{\text{sur}}$ and $\Delta t_{\text{sub}}$). The subsurface module is executed only when $t_{\text{sub}} + \Delta t_{\text{sub}} < t_{\text{sur}}$, where $\Delta t_{\text{sub}}$ is bounded by a user-defined maximum time step, $\Delta t_{\text{sub,max}}$. Thus, the user can control the time lag of the subsurface module by adjusting $\Delta t_{\text{sub,max}}$. The surface time step, $\Delta t_{\text{sur}}$, is automatically adjusted and constrained by the CFL condition.

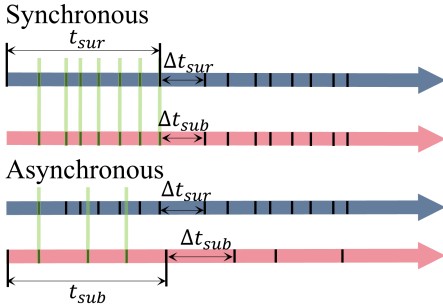

**Figure 3.** The temporal coupling strategy between subdomains.

## 2.3 Parallelization

SERGHEI-SWE-RE builds upon SERGHEI-SWE and SERGHEI-RE, achieving portability through the Kokkos framework. For detailed explanation of portability implementation in each module, refer to Caviedes-Voullieme et al. (2023) and Li et al. (2025). When using MPI for distributed memory parallelization, the domain decomposition is consistent across both surface and subsurface domains, as illustrated in Fig. 4. The computations within each partitioned subdomain are identical. As noted in Li et al. (2025), to ensure the modeling of surface hydrodynamics in all subdomains, the domain decomposition is performed only along the $x$ and $y$ directions, with user-defined partition numbers.

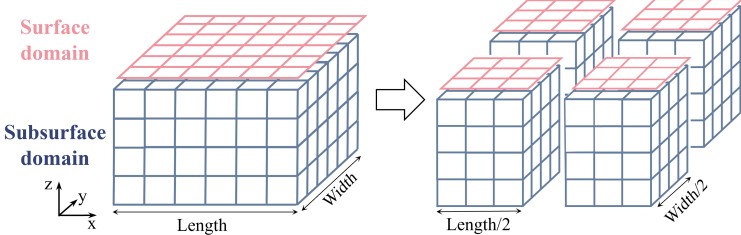

**Figure 4.** The schematic of SERGHEI-SWE-RE domain decomposition. The surface and subsurface domain decomposition is consistent, with user-defined number of partitions in $x$ and $y$ directions.





# 3 Model Verification

In this section, we evaluate the performance of the SERGHEI-SWE-RE model through five verification cases. Case 1 and Case 2 assess the model's performance under static and dynamic surface water level conditions, respectively, validating its C-property and dynamic performance. Case 3 examines a tilted v-catchment scenario to evaluate the accuracy of infiltration and surface runoff calculations, as benchmarked in Maxwell et al. (2014) and Kollet et al. (2017). Case 4 simulates rainfall on a sloped domain, accounting for both vertical infiltration and lateral subsurface flow motion (Beegum et al., 2018; Brandhorst et al., 2021). Case 5 is the superslab benchmark problem reported in Kollet and Maxwell (2006) and Kollet et al. (2017), featuring heterogeneous soil properties to represent more realistic geological conditions. The following sections provide detailed information on the cases and simulation results.

## 3.1 Case 1: Stationary Simulation

This case study examines a square domain with a bump topography. The domain measures 260 m on each side, with a uniform background bottom elevation ($z$) of 1 m and impermeable boundaries on all sides. At the point of the domain (90 m, 90 m), a 3D axisymmetric bump rises from the base, following a parabolic profile described by $z(x,y) = bump_{max} \times (1 - r^2/R^2)$, where $r = \sqrt{(x-x_0)^2 + (y-y_0)^2}$. The bump has a circular base with radius $R = 100$ m and reaches a maximum height ($bump_{max}$) of 10 m above the base level. The soil properties are uniform throughout the domain: $\alpha = 6m^{-1}$, $n = 2$, $K_s = 1.5096 m/s$, $\theta_s = 0.3$, $\theta_r = 0.08$, $\phi = 0.3$, and $S_s = 1.0 \times 10^{-5} m^{-1}$. Initially, both the surface water level and water table are set at an elevation of 6 m. The simulation runs for 1000 s. As shown in Figure 5, the water depth ($h$) remains unchanged between the start and the end of the simulation. Further quantitative examination (not shown) confirms that both the free surface elevation and subsurface pressure fields maintain their initial states during the entire simulation as expected, indicating that the C-property is satisfied.

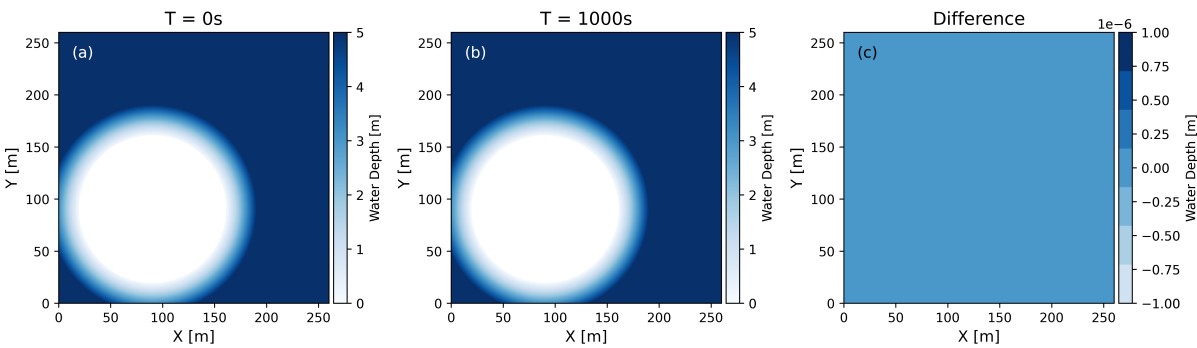

**Figure 5.** Simulated water depth for Case 1: (a) T=0 s, (b) T=1000 s, and (c) the absolute difference between (a) and (b).



## 3.2 Case 2: Fluctuating Free Surface

This test case maintains the same domain configuration as Case 1, except that a fluctuating surface water level is enforced at the inlet boundary ($x$=260 m), following the hydrograph shown in Figure 6. As the inlet water level varies, both the interior water

level and inundation area change accordingly, causing variations in the water table. Figure 7 illustrates the spatiotemporall variations of surface water depth across the domain. Figure 8 presents the simulation results of the surface water level and the water table at different times along the profile $y = 90$ m. The continuity between the free surface water level and water table is consistently maintained throughout the simulation, demonstrating the reliability of the coupled surface-subsurface flow simulation.

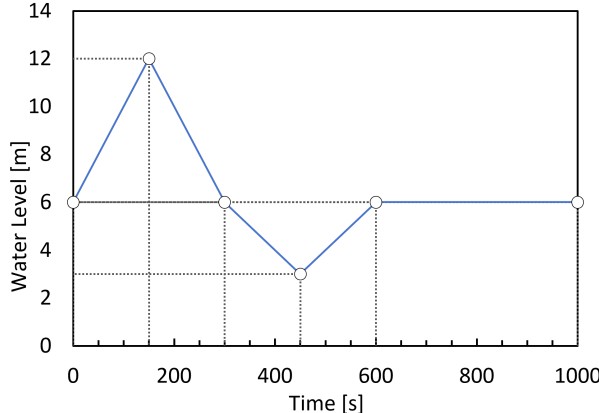

**Figure 6.** Inlet hydrograph for Case 2.





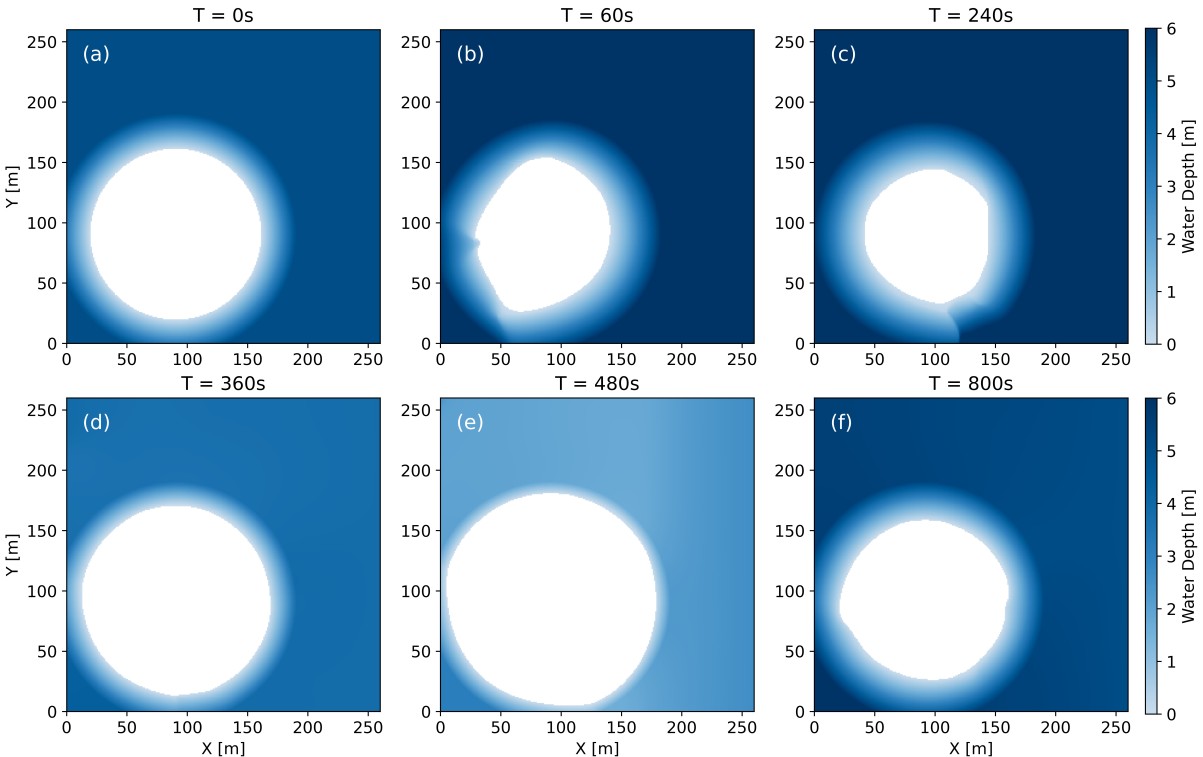

**Figure 7.** The water depth simulation results for Case 2 at (a) T=0 s, (b) T=60 s, (c) T=240 s, (d) T=360 s, (e) T=480 s, and (f) T=800 s on the transect of $y = 90$ m.




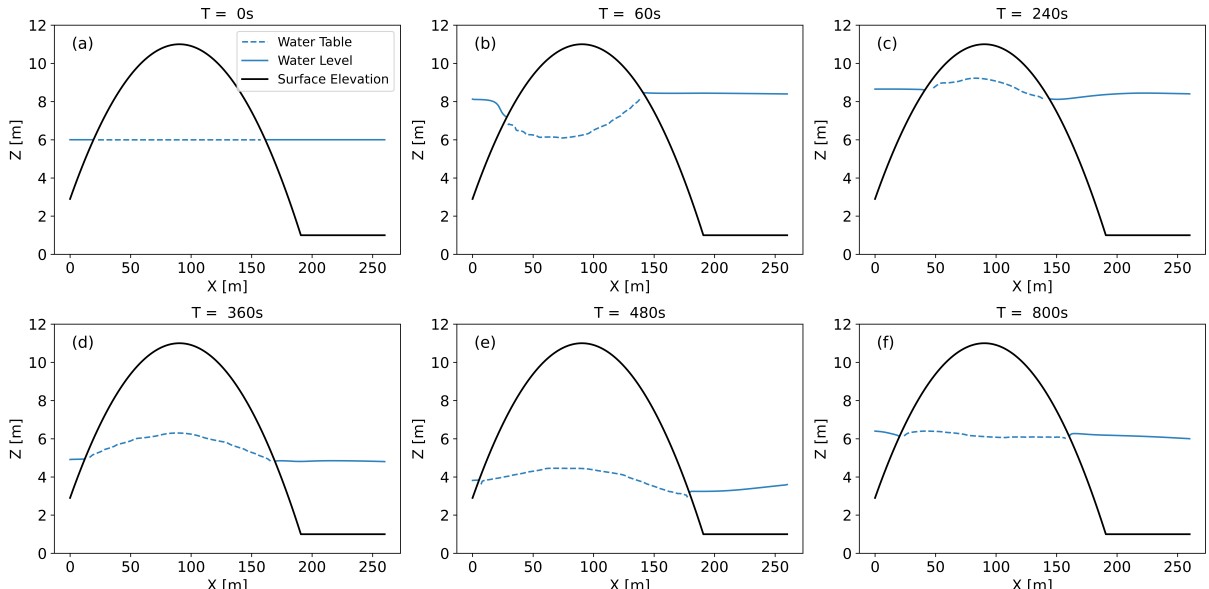

**Figure 8.** The water table and surface water level simulation results for Case 2 at (a) T=0 s, (b) T=60 s, (c) T=240 s, (d) T=360 s, (e) T=480 s, and (f) T=800 s.

## 3.3 Case 3: Tilted v-Catchment

The tilted v-catchment has been used in the model comparison study of Kollet et al. (2017) for benchmarking coupled surface-subsurface flow simulations. The configuration of the model domain is shown in Figure 9. The grid resolution is 1 m in both $x$ and $y$ directions. The Manning's coefficient is set to $\mu_1 = 1.74 \times 10^{-4} h/m^{1/3}$ for the slopes and $\mu_2 = 1.74 \times 10^{-3} h/m^{1/3}$ for the bottom channel. The subsurface domain depth is 5 m and is divided uniformly into 25 layers, with a water table 2 m below the surface as the initial condition. The bottom of the subsurface domain is impermeable. The soil is homogeneous, with $K_s = 2.78 \times 10^{-3} m/s$, $\alpha = 6m^{-1}$, $n = 2$, $\theta_s = 0.4$, $\theta_r = 0.008$, $\phi = 0.4$, and $S_s = 1.0 \times 10^{-5} m^{-1}$.





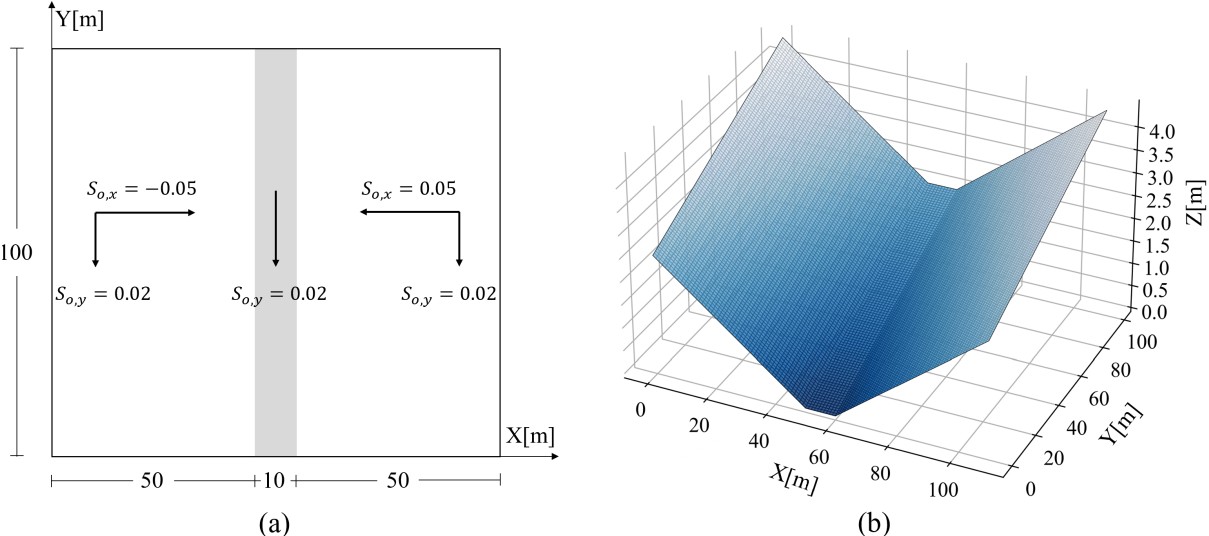

**Figure 9.** The dimension of the model domain for the tilted v-catchment simulation, modified from Kollet et al. (2017).

Two scenarios are established, both with a duration of 120 h. In Scenario 1, the subsurface water moves downstream under gravity and then forms a return flow to the surface domain, resulting in surface runoff at the channel outlet. In Scenario 2, an idealized rainfall with an intensity of 100 mm/h is added during the first 20 h, followed by a 100 h recession period. In this case study, the simulation results of SERGHEI-SWE-RE are compared with those of four existing models (ParFlow, CATHY, HGS, and Cast3M) as reported in Kollet et al. (2017).

The simulation results of the discharge at the channel outlet are depicted in Figure 10. In Scenario 1, the discharge onset time is about T=15 h for SERGHEI-SWE-RE, which aligns well with the other four models. In terms of the discharge hydrograph, SERGHEI-SWE-RE generally matches the results of the other models, with the closest resemblance to ParFlow. In Scenario 2, SERGHEI-SWE-RE has good agreement with all other models in terms of the modeled outlet discharge.



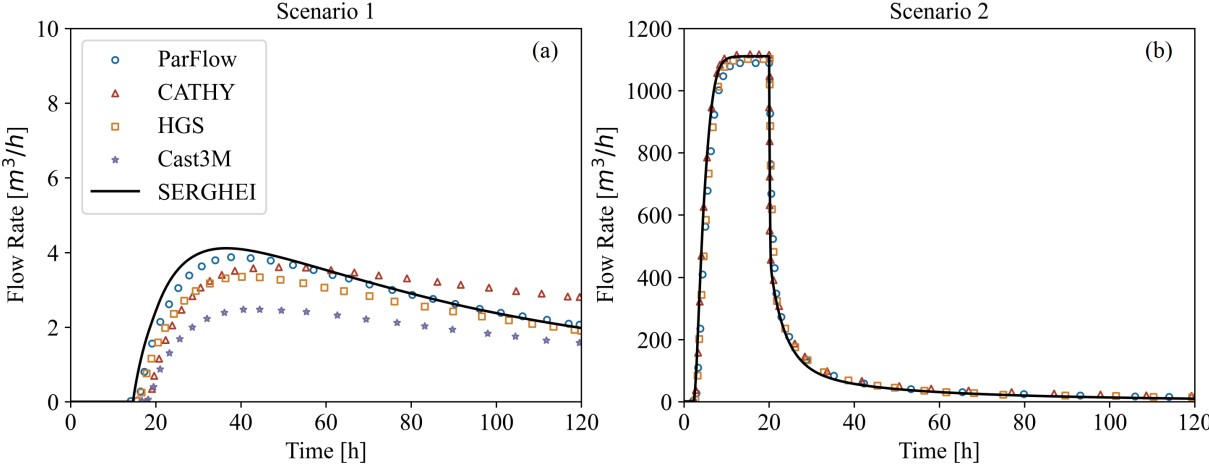

**Figure 10.** The simulated discharge at the outlets of (a) Scenario 1 (no rainfall) and (b) Scenario 2 (with rainfall).

## 3.4 Case 4: Lateral Flow

Case 4 models rainfall-infiltration into an inclined slope with open lateral boundaries. The original model domain is 3D, but can be reduced to a 2D $x$-$z$ simulation due to symmetry. Since the rainfall rate is less than the saturated hydraulic conductivity, no ponding occurs. Thus, this test case can be modeled with a Richards solver only and has been used for verifying SERGHEI-RE (Li et al., 2025). Herein, we reproduce the simulation result by activating the SWE solver, which means that precipitation is first added to the surface domain and then infiltrates into the subsurface through the SSE computation capabilities. As depicted in Figure 11(a), the 2D model domain is 4000 m in length and 15 m in height. The water table elevation is fixed at 7 m and 0.9 m on the left and right boundaries, respectively. Hydrostatic pressure distribution is applied below the water table and a constant pressure head of -1.25 m is enforced above the water table. The subsurface domain is divided into 60 layers with uniform grid resolution and homogeneous soil properties: $K_s = 5.78 \times 10^{-4} m/s$, $\alpha = 1.65 m^{-1}$, $n = 2$, $\theta_s = 0.45$, $\theta_r = 0.1$, $\phi = 0.45$ and $S_s = 1.0 \times 10^{-5} m^{-1}$. The simulation spans a 5-y period of continuous rainfall, with a consistent intensity maintained each year. Monthly variations in rainfall intensity are presented in Figure 11(b).



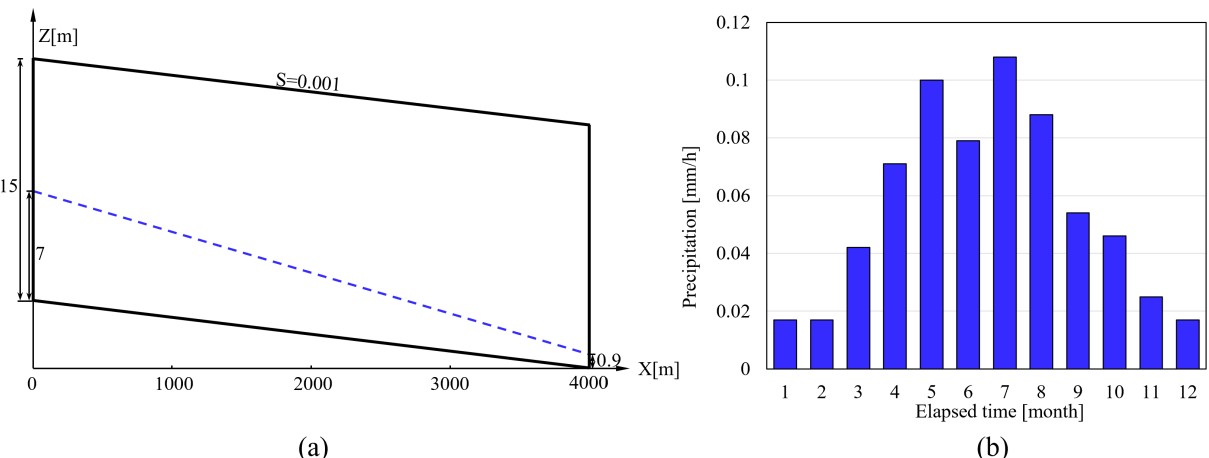

**Figure 11.** (a) the model domain configuration of the 2D case and (b) the monthly rainfall intensity.

The results of the SERGHEI-SWE-RE calculations are compared with those of the HYDRUS model and SERGHEI-RE as described in Beegum et al. (2018); Li et al. (2025). Given that the entire rainfall infiltrates the subsurface without generating

surface ponding or runoff, SERGHEI-SWE-RE and SERGHEI-RE are expected to produce identical results. Figure 12 presents the water table distribution along the $x$ direction at the end of the simulation and its variation at the domain center ($x$=2000 m). The results demonstrate excellent agreement between SERGHEI-SWE-RE and SERGHEI-RE, while being slightly higher than that of the HYDRUS model. The consistent simulation of lateral seepage flow dynamics validates that SERGHEI-SWE-RE can accurately simulate rainfall that completely infiltrates from the surface and reliably characterize groundwater table dynamics

in domains with lateral water flow.

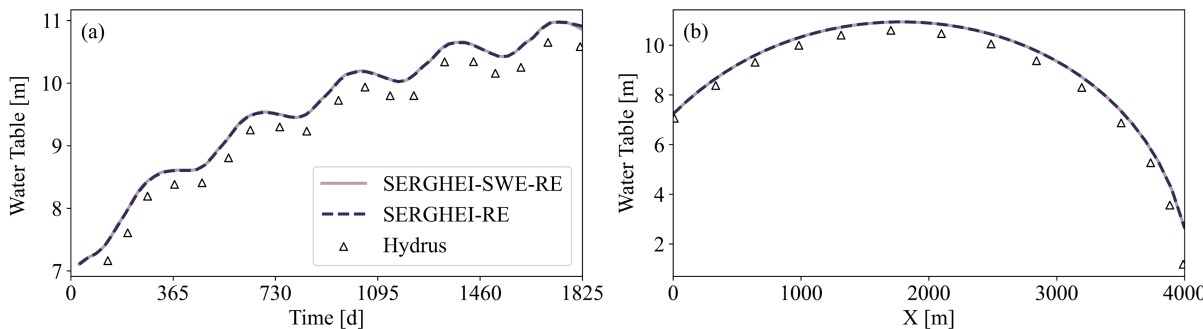

**Figure 12.** Water table (a) at the domain center and (b) at the end of the simulation along the $x$-direction.

### 3.5 Case 5: Heterogeneous Superslab

Case 5 considers the superslab benchmark problem reported in Kollet et al. (2017), which consists of rainfall-runoff simulation on a sloped plane. The plane has a length of 100 m in the $x$-direction and a slope of 0.1 (Figure 13). The Manning's roughness




coefficient is $\mu = 1.0 \times 10^{-6} h/m^{1/3}$. Two soil slabs (slab 1 and slab 2) are present in the subsurface domain with significantly
lower conductivities ($K_s$) than the background soil. The properties of the soils are detailed in Table 1. The perimeter and
bottom of the subsurface domain are impermeable. The simulation period is 12 h, with a constant rainfall intensity of 50 mm/h
applied for the initial 3 h.

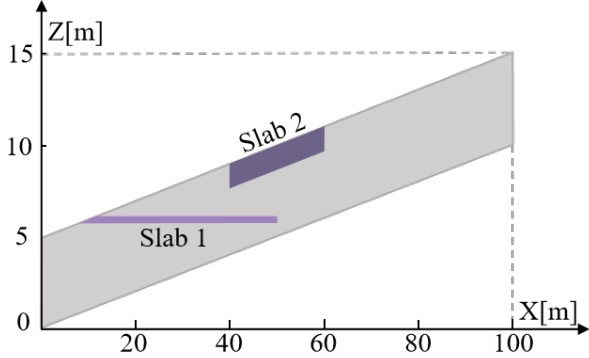

**Figure 13.** The problem domain configuration of the slab case with three soil types.

**Table 1.** The specific parameters for soil types.

| Parameter | Background | Slab 1 | Slab 2 |
|---|---|---|---|
| Lateral extension in $x$ [m] | 0–100 | 8–50 | 40–60 |
| Vertical extension in $z$ [m] | 5 m below land surface | 5.8–6.2 | 1.3 m below land surface |
| $K_s$ [m/h] | 10 | 0.025 | 0.001 |
| $\phi$ | 0.1 | 0.1 | 0.1 |
| $S_s$ [m$^{-1}$] | $1.0 \times 10^{-5}$ | $1.0 \times 10^{-5}$ | $1.0 \times 10^{-5}$ |
| $n$ | 2.0 | 3.0 | 3.0 |
| $\alpha$ [m$^{-1}$] | 6.0 | 1.0 | 1.0 |
| $\theta_r$ | 0.02 | 0.03 | 0.03 |
| $\theta_s$ | 0.1 | 0.1 | 0.1 |

The SERGHEI-SWE-RE simulation result is compared with with five existing models (ATS, CATHY, HGS, Cast3M, and
ParFlow) reported in Kollet et al. (2017). For this benchmark problem, ParFlow produces very similar discharge hydrograph
and soil saturation profiles to ATS, so it is not displayed herein for simplicity. Furthermore, the simulation results obtained
from synchronous and asynchronous coupling approaches are all presented. In synchronous coupling approach, the surface
and subsurface solvers are advanced with the same time step. The surface time step ($\Delta t_{\text{sur}}$) typically ranges from 0.4 to 0.55
s for this case. In asynchronous coupling simulations, the maximum time steps of the RE solver ($\Delta t_{\text{sub,max}}$) is set to 1 s
(SERGHEI-asy-1s) and 2 s (SERGHEI-asy-2s), respectively.





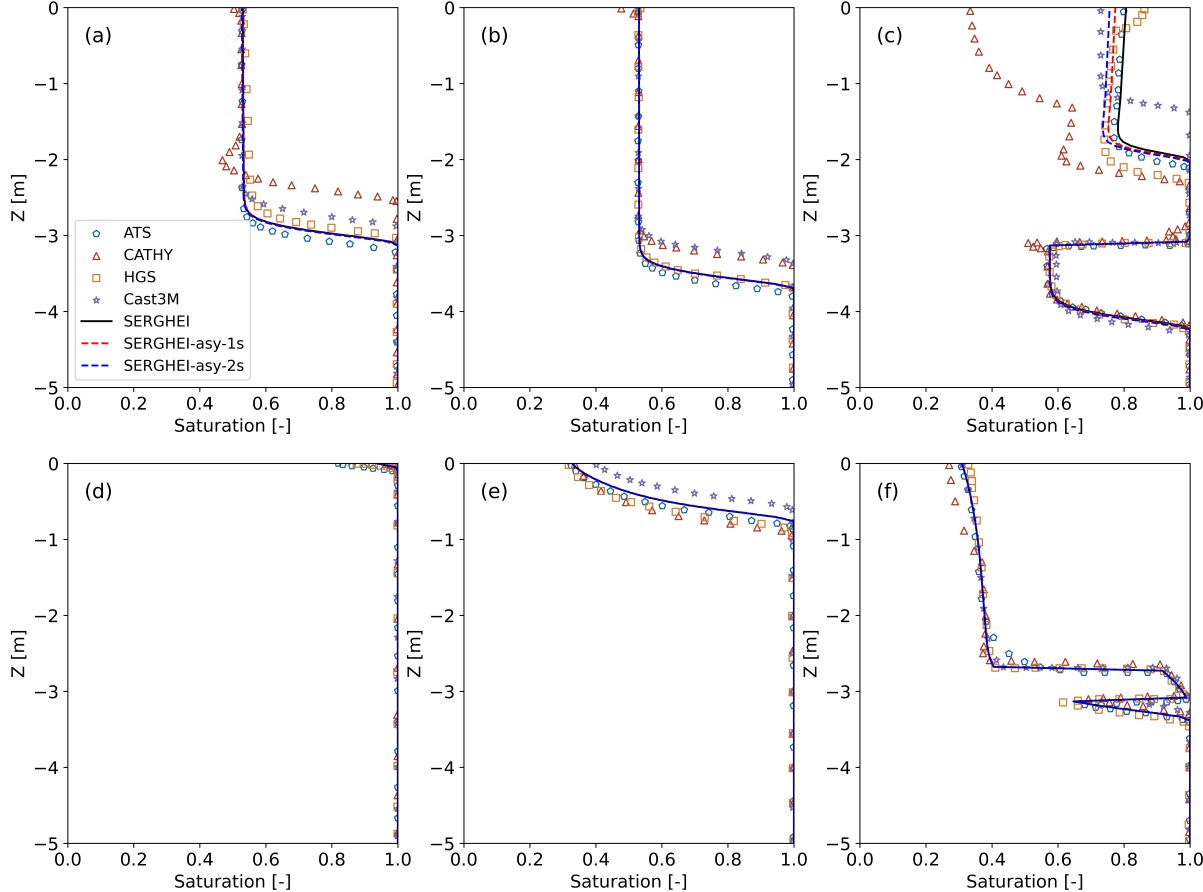

**Figure 14.** The simulated soil saturation profiles for (a)-(c) T=3 h and (d)-(e) T=6 h at $X = 0$, 8, 40 m. SERGHEI represents the synchronous calculation; SERGHEI-asy-1s and SERGHEI-asy-2s represent the asynchronous calculation with maximum time steps of 1s and 2s, respectively

Figure 14 shows simulated soil saturation profiles at the outlet ($x$=0 m), the edge of slab 1 ($x$=8 m) and the edge of slab 2 ($x$=40 m), respectively. At all three locations, SERGHEI-SWE-RE remains within the envelope of existing models, and aligns well with the ATS and HGS profiles at $x$=0 m and 8 m. $x$=40 m is more challenging as the reference models produce divergent saturation profiles (Fig. 14(c)). Nevertheless, SERGHEI-SWE-RE stays close to most of the reported models, indicating that it is capable of capturing soil water dynamics in heterogeneous soil media.

Figure 15 presents the simulated outlet discharge and surface water volume. As shown in Fig. 15(a), the discharge hydrograph of SERGHEI-SWE-RE aligns well with ATS and Cast3M. In contrast, notable discrepancies are observed when compared to the results from CATHY and HGS. For the surface ponding volume, Fig. 15(b) shows a significant difference between SERGHEI-SWE-RE and the other models, with SERGHEI-SWE-RE consistently producing higher ponding volumes. In this case, surface ponding primarily arises from the combined effect of direct rainfall and runoff from the upslope. To further





investigate the source of the discrepancy, a separate simulation of rainfall-runoff on slab 2 is performed, with infiltration disabled and upslope/downslope sections truncated. The resulting ponding volume, shown as the blue dashed line in Fig. 14(b), illustrates that the discrepancy of SERGHEI-SWE-RE persists. Clearly, this discrepancy originates from the SWE solver rather than the SSE coupling. SERGHEI-SWE-RE solves the fully dynamic 2D SWE. In contrast, the benchmark models employ simplified approximations, such as the kinematic wave equations (ParFlow and CATHY) or the diffusive wave equations (ATS,

HGS and Cast3M). Consistent with this, Caviedes-Voullième et al. (2020) reported that the simplified model systematically underestimates water depths compared to the full SWE model in rainfall-runoff simulations. Therefore, the model discrepancy in Fig. 15(b) does not undermine the validation of SERGHEI-SWE-RE for reliable surface–subsurface exchange simulations.

    As shown in Figures 14 and 15, although the synchronous and asynchronous coupling results of SERGHEI-SWE-RE are slightly different, they both achieve satisfactory agreement with existing models in simulating soil saturation and outlet dis-

charge. Moreover, the wall clock times for synchronous and asynchronous coupling simulations are 29.75 (synchronous), 20.3 (SERGHEI-asy-1s) and 11.97 (SERGHEI-asy-2s) minutes, respectively (all performed on a Nvidia L40 GPU). Using asynchronous coupling reduces computation times by 31.76% and 59.77% for SERGHEI-asy-1s and SERGHEI-asy-2s scenarios. This implies that asynchronous surface-subsurface coupling of SERGHEI-SWE-RE ensures accurate simulation results while saving computational cost.

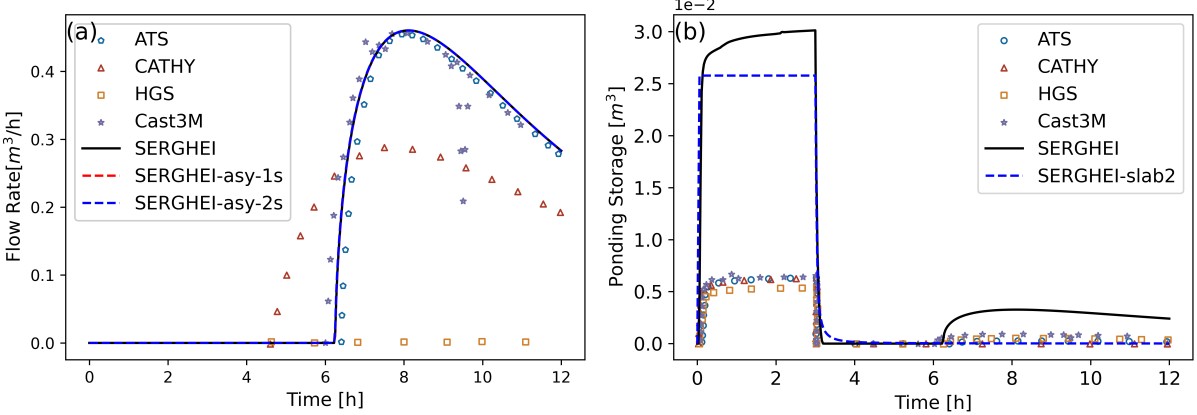

**Figure 15.** The temporal variations in (a) outlet discharge and (b) surface water volume. SERGHEI-asy represents the scenarios with asynchronous surface-subsurface coupling. SERGHEI-slab2 is a customized SWE-only simulation on the slab 2, with the upslope and downslope sections truncated.

# 4  Scalability Analysis

## 4.1  Strong Scaling Behavior

We evaluate the scalability and performance portability of SERGHEI-SWE-RE through large-scale simulations of Lake Taihu, located in the Yangtze River Delta, encompassing coupled surface-subsurface flow dynamics. In collaboration with the Taihu





Basin Monitoring Center of Hydrology and Water Resources, we are developing a coupled surface-subsurface flow model for

Lake Taihu using SERGHEI-SWE-RE. Although the complete model is not available yet, the semi-finished model is adequate

for conducting scaling tests, which involve fairly complex topography and multiple realistic boundary conditions.

The model domain encompasses an area of approximately 5500 $\text{km}^2$ and extends to a depth of 30 m. Vertically, the sub-

surface domain is discretized into 10 layers with variable grid resolutions ranging from 1 m to 7.45 m. Horizontally, two grid

resolutions are used ($\Delta x$ = 200 m and 50 m), resulting in a total of 1,521,828 and 24,349,248 grid cells (including both surface

and subsurface domains), respectively. To construct the Lake Taihu model, we integrate topographic data provided by the Taihu

Lake Authority for the submerged lake bottom with a freely available 30-meter resolution digital elevation model (DEM) data

for the surrounding area. We merge the 30 major rivers that connect to the lake into 9 artificial rivers. We also merge the inflow

and outflow rate data provided by the Taihu Lake Authority, and distribute the flow rates to the 9 rivers, making sure that the

total inflow and outflow of the lake are close to the real conditions. We delineate 15 sections of subsurface boundaries where a

prescribed water table elevation is enforced. Due to data limitations, the soil properties across the entire study area are assumed

to be uniform. Evaporation and rainfall are neglected for simplicity. A wind subroutine is implemented into SERGHEI-SWE

to simulate the wind-driven oscillation of the lake water level, see Eq. (5):

$$\tau_x = C_\text{D} \rho_a \cos\omega \left(u_w - |u|\cos\beta\right)^2$$
$$\tau_y = C_\text{D} \rho_a \sin\omega \left(u_w - |u|\cos\beta\right)^2 \tag{5}$$

where, $\tau_x$ and $\tau_y$ are the wind stresses in x and y directions $[ML^{-1}T^{-2}]$, $\rho_a$ is air density $[ML^{-3}]$, $u_w$ is wind speed $[LT^{-1}]$,

$|u|$ is water speed $[LT^{-1}]$, $\omega$ is the angle from the x axis to the wind direction, $\beta$ is the angle between the wind direction and

the water flow direction, $C_\text{D}$ is the wind drag coefficient, and is set to 0.0013 in this simulation. To avoid instability, wind stress

is deactivated if the water depth is less than 0.1 m (Li and Hodges, 2019).

For the scaling tests, Lake Taihu is simulated for 10 days on three HPC facilities: (i) a small cluster at the high-performance

computing center of Tongji University (named TJ HPC hereafter), where each CPU node is equipped with an Intel Xeon Max

9468 processor (3.5 GHz, 48 cores, 96 threads), and each GPU node is equipped with an Nvidia L40 GPU (48 GB memory,

864 GB/s bandwidth, 18176 CUDA cores), (ii) the LISE HPC system of the German National High Performance Computing

(NHR) Alliance at Zuse Institute Berlin, Germany. Each computation node of LISE is equipped with two units of Intel Xeon

Platinum 9242 processors (2.30 GHz, 48 cores, 96 threads), and (iii) the JUWELS booster system at the Jülich Supercomputing

Center. Each JUWELS booster node contains 4 Nvidia A100 Tensor Core GPU cards (80 GB memory, 1935 GB/s bandwidth).

Figure 16 illustrates the simulation time and speedup achieved on the TJ HPC ($\Delta x$ =200 m). It is evident that, on a single

CPU node, the scaling is nearly ideal up to 16 threads. However, starting from 32 threads, the scaling begins to deteriorate.

A closer examination reveals that the SWE solver demonstrates poorer scaling compared to the RE solver. The SWE solver

exhibits poor scaling due to: (i) lower computational workload due to fewer surface grid cells (than the subsurface domain),

(ii) lower computational intensity per grid cell, (iii) higher communication-to-computation ratio, and (iv) load imbalance in

boundary condition processing. Indeed, even poorer scaling is observed for the computation of the boundary conditions (BC),

because the number of boundary cells is much fewer than the number of interior cells. A similar trend is observed on the GPU,





where the speedup of the coupled model exceeds 128, but the speedup of the SWE solver is only around 64, and the speedup of the BC computation is less than 8. It is important to note that the total computational workloads for BC and the SWE solver are significantly smaller than that for the RE solver. Although the BC and the SWE solver are less efficient with an increasing number of threads, the overall scaling of the coupled simulation is predominantly influenced by the scaling of the RE solver.

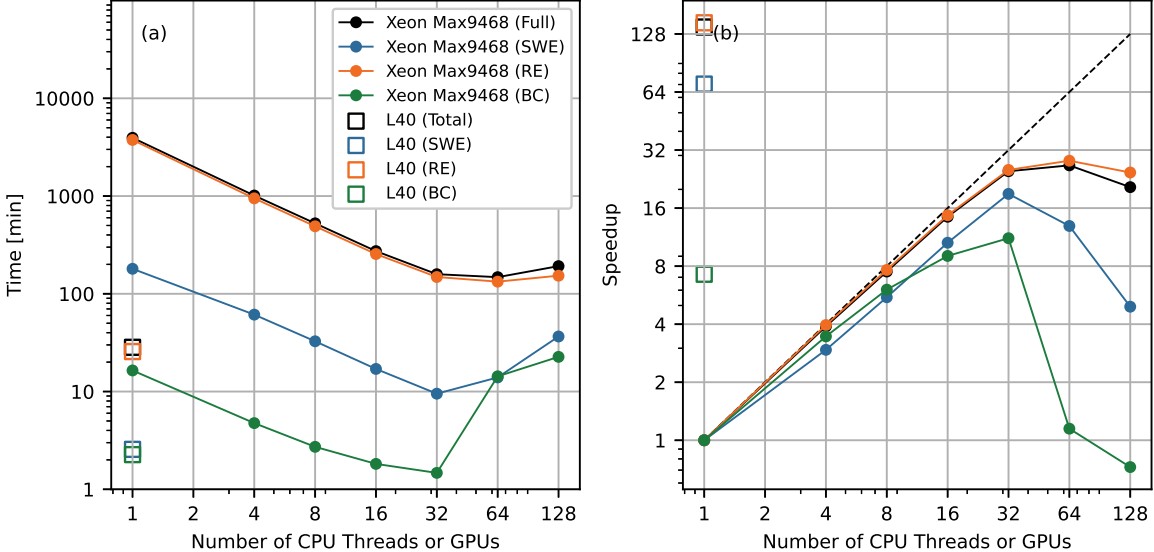

**Figure 16.** (a) Simulation time and (b) speedup for the Lake Taihu simulation ($\Delta x =$200 m) on the TJ HPC. Full, SWE, RE, and BC in the legends represent the simulation time and the speedup of the entire simulation, the SWE solver, the RE solver and the computation of the boundary condition, respectively. Note that the horizontal axis is the number of CPU threads (i.e., the number of CPUs times the number of threads used per CPU) or the number of GPUs. The dashed line represent the ideal scaling

Figure 17 shows the simulation time and speedup on the LISE HPC (with $\Delta x =$50 m). The scalability evaluation reveals that both the SWE and RE solvers maintain favorable scalability until deterioration occurs at 64 CPU nodes. In particular, the RE solver exhibits superior scalability compared to the SWE solver, a finding consistent with the observations in Figure 16. This phenomenon, previously analyzed in relation to Figure 16, can be attributed to the significantly higher count of grid cells in subsurface computations relative to surface grids. A comparative analysis between Figures 16 and Figure 17 indicates that LISE HPC achieves better scalability than TJ HPC, especially for the SWE solver, which is primarily due to the substantial increase in computational grids (from 1,521,828 grids at $\Delta x =$200 m to 24,349,248 grids at $\Delta x =$50 m).





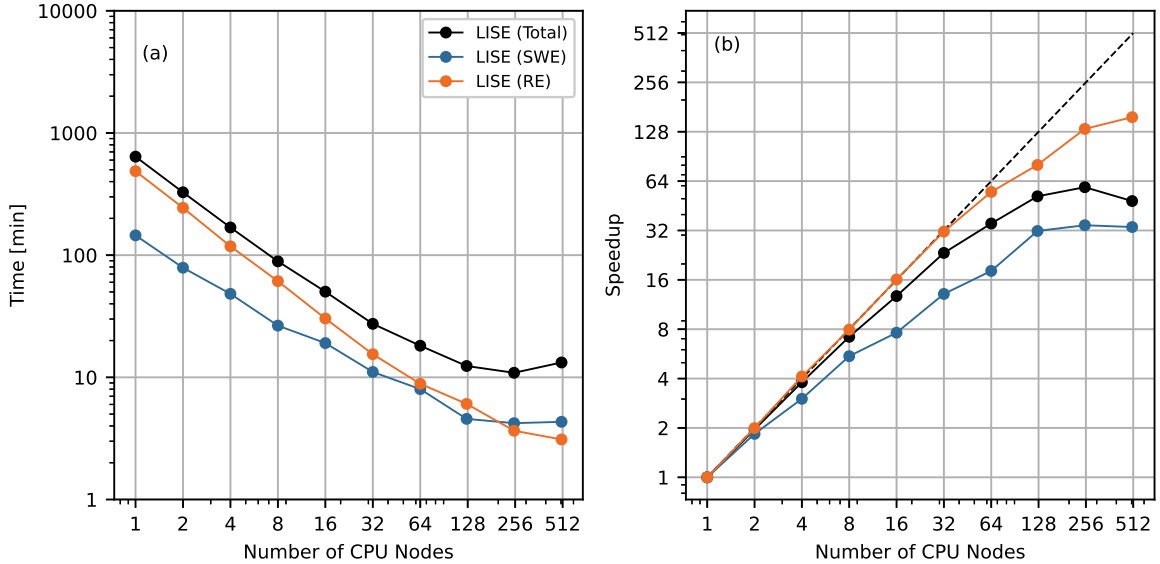

**Figure 17.** (a) Simulation time and (b) speedup for the Lake Taihu simulation ($\Delta x =$ 50 m) on the LISE HPC with Intel Xeon Platinum 9242 processors. Full, SWE and RE in the legends represent the simulation time and the speedup of the entire simulation, the SWE solver and the RE solver, respectively. The dashed line represent the ideal scaling

Figure 18 presents the GPU acceleration performance on the JUWELS system. Analysis reveals distinct computational char-
acteristics between solvers: the RE solver demonstrates superlinear speedup (2-8 GPUs) with significantly reduced computation
time, while the SWE solver shows minimal improvement and even exhibits increased simulation time at 256 GPUs. Again, this
performance divergence stems from the different number of grid cells in the surface and subsurface domains. The surface do-
main contains only about 2 million grid cells, which is insufficient to fully utilize the computational power of multiple Nvidia
A100 GPUs. Figure 18 also illustrates that as the number of GPUs increases, the MPI communication time (including both
surface and subsurface domains) approaches the SWE solver computation time, indicating that MPI communication overhead
becomes a bottleneck for scaling with more GPUs. For coupled surface-subsurface flow simulations, the number of subsurface
cells equals the number of surface cells multiplied by the number of subsurface layers. Consequently, the optimal paralleliza-
tion configuration for the RE solver becomes inefficient for the SWE solver due to significant disparities in the computational
workloads. It indicates that the simple consistent surface-subsurface cell mapping and domain decomposition strategy (Fig. 3
and 4) might be sub-optimal for large-scale parallelization. Future enhancements to SERGHEI-SWE-RE should incorporate
either (i) different mesh resolutions for the SWE and RE solvers or (ii) workload-based domain decomposition schemes to
improve scaling performance.

Although the SWE solver scales poorly on multi-GPU system, the overall speedup is acceptable up to 64 GPU nodes
because the computational load of the SWE solver is much less than the RE solver. Moreover, compared to Fig. 16, multi-GPU
computing remains faster than multi-thread CPU computing, despite the former consists of 16 times more grid cells ($\Delta x =$ 50





m and 200 m respectively). The scaling test results (Fig. 16 to 18) illustrate that SERGHEI-SWE-RE is capable of performing large-scale, coupled surface-subsurface flow simulations on a variety of HPC systems.

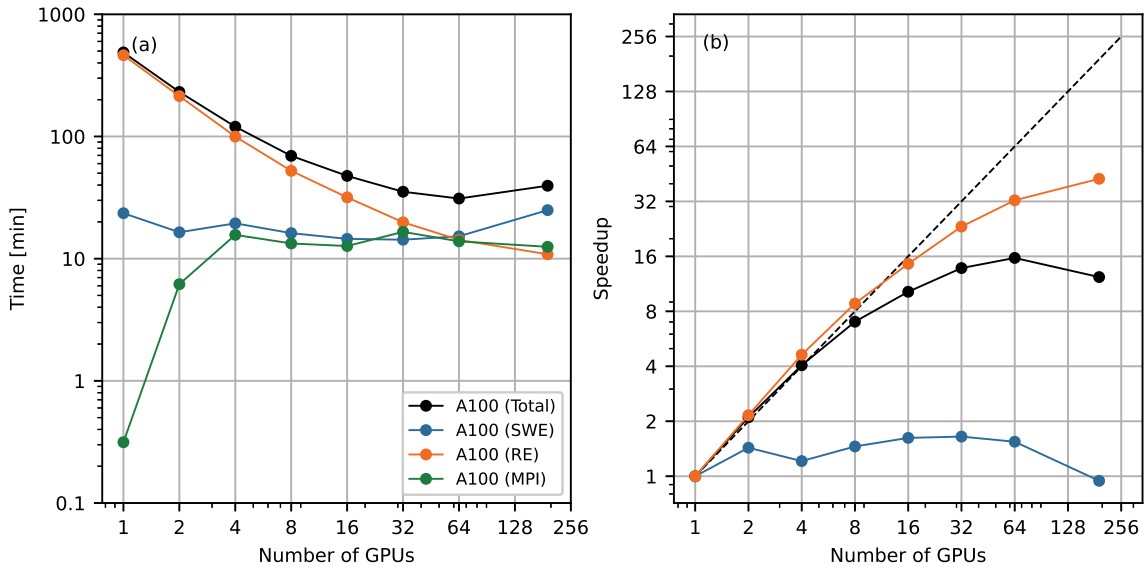

**Figure 18.** (a) Simulation time and (b) speedup for the Lake Taihu simulation ($\Delta x =$50 m) on the JUWELS Booster with Nvidia A100 GPUs. Full, SWE, RE and MPI in the legends represent the simulation time and the speedup of the entire simulation, the SWE solver, the RE solver, and the MPI communication (for both SWE and RE solvers), respectively. The dashed line represent the ideal scaling

## 4.2 Weak Scaling Behavior

A second scaling analysis is performed to demonstrate the weak scaling behavior of SERGHEI-SWE-RE. The 2D $x$-$z$ domain

of the lateral flow problem (Section 3.4) is extended and duplicated along the $y$ axis to create a 3D test case, where each vertical layer($x$-$z$) is identical. Thus, by adjusting the number of vertical layers in the $y$ direction, the dimension of the computational domain can be customized to maintain the same workload per processor, without affecting the simulation results. A similar domain configuration approach has been reported in Li et al. (2025).

The weak scaling test is conducted on the JUWELS booster system. The domain dimensions are adjusted to ensure that

each GPU node is responsible for a subdomain containing 4.8 million subsurface cells and 80 thousand surface cells. Figure 19 shows the parallel efficiency – defined as the ratio between single-node and multi-node execution times – as a function of the number of GPU nodes used. It can be seen that the parallel efficiency of the entire model slightly decreases as more GPUs are employed, but remains higher than 0.8 with 16 GPU nodes, indicating that SERGHEI-SWE-RE can efficiently handle larger problems as the available computational resources increase. The parallel efficiency of the RE solver closely matches the overall

efficiency. The SWE module exhibits more variations in the parallel efficiency, but since the simulation time for the SWE solver is relatively small compared to the RE solver, this has a negligible impact on the overall efficiency.





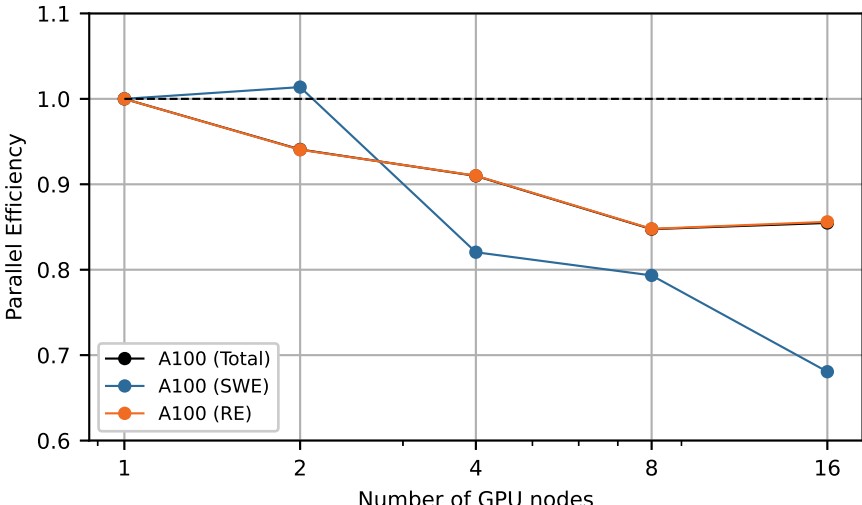

**Figure 19.** Parallel efficiency for the weak scaling tests performed with the extended lateral flow problem, on the JUWELS Booster with Nvidia A100 GPUs. Full, SWE and RE in the legends represent the efficiency of the entire simulation, the SWE solver and the RE solver, respectively. The dashed line represent the ideal efficiency.

## 5 Conclusions

In this paper, we introduce an integrated surface-subsurface hydrodynamic model, SERGHEI-SWE-RE, based on the fully dynamic 2D shallow water equation solver (SERGHEI-SWE) and the 3D Richards equation solver (SERGHEI-RE), utilizing the Kokkos framework. The coupled model employs a sequential coupling approach, offering both synchronous and asynchronous coupling modes. The coupled surface-subsurface flow exchange simulation results reach good agreement with existing models on the benchmark problems tested, encompassing both homogeneous and heterogeneous soil conditions, as well as scenarios with and without rainfall. Notably, the asynchronous coupling mode significantly improves computational efficiency while maintaining accuracy. The parallel scalability and performance portability of the model are further evaluated using a real-world case study involving over 24 million grid cells on multiple HPC systems. The results indicate that SERGHEI-SWE-RE performs efficiently across various HPC platforms with different hardware architectures. The SWE solver generally scales poorer than the RE solver due to much less computational workload. Further enhancements on scaling can be expected by improving grid partitioning strategies between the surface and the subsurface domains.

*Code and data availability.* SERGHEI is available through GitLab, at https://gitlab.com/serghei-model/serghei, under a 3-clause BSD license. Developer and user guides are available in the wiki page of the SERGHEI project. The following tools and packages are pre-requisite for compiling and running SERGHEI-SWE-RE: GCC (other C++ compilers have not been tested), OpenMPI (other MPI implementations have not been thoroughly tested), CMake, Kokkos, KokkosKernels, PnetCDF. A static version of SERGHEI-SWE-RE used for this



manuscript, and the input files of the test cases, are archived in Zenodo, with DOI: https://doi.org/10.5281/zenodo.17217612 (Li, 2025). Due to third-party restrictions, some of the Lake Taihu input data cannot be redistributed by the authors. The data are provided by Taihu

Basin Monitoring Center of Hydrology and Water Resources under internal use terms. Access requests should be made through email to the corresponding author of this manuscript.

*Author contributions.*   NZ contributed to conceptualization, methodology, software, formal analysis, visualization, and writing(original draft). ZL contributed to conceptualization, methodology, software, supervision and writing(original draft). GR contributed to software, formal analysis and writing(original draft). MMH contributed to methodology, resources, and writing(review & editing). IOX contributed to methodol-

ogy, resources and writing(review & editing). DCV contributed to conceptualization, resources, supervision and writing(review & editing).

*Competing interests.*   The authors declare no competing interests.

*Acknowledgements.*   This study is supported by the National Natural Science Foundation of China (NSFC Grant No. 42307078), the National Key R&D Program of China (2022YFC3803000), and the Fundamental Research Funds for the Central Universities (China). This work used HPC resources provided by the Center for Scientific Computing at Tongji University, China, and the German National High Performance

Computing (NHR) Alliance at Zuse Institute Berlin (ZIB), Germany. G. Rickert's work was funded as part of the sTREssE project by the German Research Foundation (DFG Grant No. 545756575). The authors gratefully acknowledge the Taihu Basin Monitoring Center of Hydrology and Water Resources for providing terrain and hydrology data of Lake Taihu.



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
