# Peer review of "High-performance coupled surface-subsurface flow simulation with SERGHEI-SWE-RE"

_EGUsphere, 2025_

## Author Comment (AC1)

**Response to reviewers**

**Reviewer 1:**

This manuscript presents SERGHEI-SWE-RE, a coupled surface–subsurface model that links a fully dynamic 2D shallow-water solver with a 3D Richards solver. Overall, the work is solid and clearly within the scope of GMD. The model is well described, the benchmark cases are appropriate, and the scaling analysis across several HPC systems is particularly useful. The asynchronous coupling strategy is also a strength of the paper, and the manuscript is generally clear and well organized. I find the manuscript suitable for publication after a set of minor revisions. My comments below are intended to improve clarity and strengthen the presentation.

**Major Comments:**

1. The sequential coupling strategy is well explained, but it would be helpful to briefly discuss its potential limitations. In particular, readers would benefit from understanding situations in which asynchronous coupling may introduce small inaccuracies, or cases where a fully coupled approach might be more appropriate. A short clarification in Section 2.2 or in the Discussion would be sufficient. For example, in Case 5 the authors test two maximum RE time steps, which affects the overall runtime almost linearly. While I am not requesting additional experiments, it would strengthen the paper to comment on how sensitive the results are to the chosen coupling time window. How does varying the subsurface time step influence accuracy and computational cost? A brief discussion of this trade-off would help contextualize the asynchronous approach.

[Response: Thank you for this valuable suggestion. we have (i) expanded the details in Section 2.2 to provide a more comprehensive description of the asynchronous coupling strategy, explicitly addressing both its advantages and limitations, (ii) added a more systematic and quantitative comparison between the synchronous and asynchronous coupling strategies in Section 3.5, including additional simulation results, and (iii) addressed the need to further refine asynchronous coupling strategy in the Conclusions. ]

Section 2.2 :SERGHEI-SWE-RE supports both synchronous and asynchronous coupling between surface and subsurface modules. In synchronous coupling, the surface and subsurface modules are executed at every global time step, defined as the smaller of their individual time steps, ensuring both solvers advance on a common timeline. In asynchronous coupling, as shown in Fig. 3, the two modules operate on their own independent timelines, denoted as $T_{swe}^p = \sum_{i=0}^p \Delta t_{swe}^i$ and $T_{re}^q = \sum_{i=0}^q \Delta t_{re}^i$, where $p$ and $q$ are the time step indices and $\Delta t_{swe}^p$ and $\Delta t_{re}^q$ are the corresponding time step sizes. Both $\Delta t_{swe}^p$ and $\Delta t_{re}^q$ vary in time. The former is restricted by the CFL condition, and the latter is adjusted based on either the maximum change in water content (in the PC scheme) or the number of linearization iterations (in the MP scheme). A user-defined upper bound, $\Delta t_{re,max}$, is enforced to further constrain $\Delta t_{re}^q$ (Li et al., 2025). The subsurface module is executed only when $T_{re} + \Delta t_{re}^q < T_{swe}$. With this approach, the SSE flux ($q_{ss}$) is computed using the current surface flow state and the previous subsurface flow state, shown as the red arrows in Fig. 3. Similarly, the latest surface water depth is sent backward to the subsurface module as boundary conditions, shown as the blue arrows in Fig. 3. Asynchronous coupling significantly enhances computational efficiency by reducing the execution times of the RE module. However, it introduces a time lag of the subsurface module as illustrated in Fig. 3. This time lag inevitably results in coupling errors that depend on both the time scale difference between surface and subsurface flow and the lag duration, an issue extensively discussed in (Li et al., 2023) and later in Section 3.5. Practically, users can tune $\Delta t_{re,max}$ to balance coupling accuracy and computational efficiency. It is, in principle, also possible to set the $\Delta t_{re}^q = N \Delta t_{swe}^p$ to prevent an excessive difference between the two time-step sizes. However, investigation of this approach, and a full investigation of the implications of synchronocity/asynchronicity is left for future work. Finally, it is worth noting that the synchronous/asynchronous coupling strategies apply to both PC and MP schemes of the RE solver. For both schemes, the surface water depth is used as a boundary condition for implicitly solving the subsurface pressure head. Consequently, the data exchange pathways between modules, shown as red and blue arrows in Fig. 3, are identical for both solution schemes.

**Table 2.** The RMSE and the reduction in the computational cost for asynchronous coupling with different $\Delta t_{re,max}$.

| Performance Metric | $\Delta t_{re,max}$ [s] | | | |
|---|---|---|---|---|
| | 1.0 | 2.0 | 3.0 | 4.0 |
| RMSE [-] | 0.0041 | 0.0124 | 0.0203 | 0.0265 |
| Cost Reduction [%] | 29.6 | 58.2 | 69.1 | 74.77 |

[Figure]

**Figure 3.** Schematic of the asynchronous coupling strategy between surface and subsurface modules over time. The red arrows denote the transfer of the SSE flux ($q_{ss}$) as a volumetric source/sink term to the surface domain. The blue arrows denote the transfer of the surface water depth as the boundary condition to the subsurface domain.

Section 3.5 : Table 2 evaluates the performance of asynchronous coupling by comparing the root mean square error (RMSE) of the saturation profiles for $T = 3$ h at $X = 40$ m against the synchronous simulation benchmark. The improvements in computational efficiency are also reported. As expected, while the simulation time reduces as $\Delta t_{re,max}$ increases, the RMSE also rises. However, for this test case, the RMSE remains relatively low and the saturation profiles for different $\Delta t_{re,max}$ values are nearly indistinguishable (Fig. 14). The reason is that for this rainfall-runoff scenario, the surface flow field is relatively stable. In contrast, under scenarios with highly dynamic surface flow, such as tidal oscillations, asynchronous coupling may lead to non-negligible errors as demonstrated in Li et al. (2023). The optional asynchronous coupling strategy implemented in SERGHEI-SWE-RE thus provide the flexibility that allows users to balance computational efficiency and accuracy for different simulation scenarios. A full investigation of the possible implications of asynchronicity under a set of diverse problems and conditions warrants a study in itself, and sets clear future work to be carried out. In such investigation it will be important not only to document the error and computational efficiency behaviours in response to asynchronicity, but also to explore potential heuristics to optimise this tradeoff.

Section 5: The tests here show that the asynchronous coupling strategy seems to be computationally efficient, but may introduce small errors due to lags in the coupling relative to a synchronous coupling. Future work must explore fully the implications and tradeoffs of this strategy, and also account for the effect that this may have in computational resource usage and load balance given that the resource demands and scalability of the SWE and RE solvers is different.

2. One overall question the scalability discussion does not fully address is the performance impact introduced by the coupling itself. In other words, how much does the integration of the SWE and RE solvers degrade performance compared to running the two components independently? A brief quantitative or qualitative assessment of this overhead, whether due to synchronization, data exchange, or load imbalance, would help readers better understand the true cost of coupling and the efficiency of the current implementation.

[Response: We thank the reviewer for highlighting this important point. To address this concern, we have performed additional simulations to compare the

computational performance of SERGHEI-SWE-RE and SERGHEI-RE using the same benchmark problem. The results are presented and discussed in a new section, Section 4.3: ]

Section 4.3 : The coupling between the SWE and RE modules may introduce additional computational overhead that affects scalability. An additional scaling test is conducted to compare the performance of the coupled model and the standalone SERGHEI-RE module. The problem setup is identical to the lateral flow test (Section 3.4), in which rainfall infiltrates into the subsurface without causing surface runoff. Section 3.4 has demonstrated that SERGHEI-RE and SERGHEI-SWE-RE produce indistinguishable water table evolution. However, in SERGHEI-RE, rainfall is applied directly to the subsurface domain as a flux boundary condition, whereas in SERGHEI-SWE-RE, rainfall is first applied to the surface domain, inducing ponding before infiltrating into the subsurface through surface–subsurface exchange computations. To understand the scaling behaviors of these two approaches, the original 3D model domain is used, which spans 4000 m × 5000 m in the horizontal directions and 15 m in the vertical direction. The domain is discretized with uniform grid spacings of $\Delta x = \Delta y = 5m$ and $\Delta z = 1.5m$, resulting in a total of 8 million grid cells. Fig. 20 shows the simulation times and speedups of the two approaches as the number of CPU threads increases. The SERGHEI-SWE-RE and SERGHEI-RE models exhibit negligible difference in terms of both the computational time and the speedup. This behavior can be attributed to the relatively small number of surface grids cells (as discussed in Section 4.1) and the absence of surface runoff generation in this case. These results indicate that, for this particular test case, surface–subsurface coupling does not degrade model scalability.

It should be noted, however, that this conclusion applies only to this specific test scenario. In a broader context, surface-subsurface coupling is not expected to alter the scalability of the subsurface module because the SERGHEI-RE solver treats surface flow as a boundary condition (Section 2), which must be defined regardless whether SERGHEI-SWE is activated or not. Conversely, the surface module performs an additional parallel loop over all surface grid cells to incorporate the exchange flux as a source/sink term in the mass conservation equation. Thus, if SERGHEI-SWE scales poorly (as shown in Section 4.1 on JUWELS), surface-subsurface coupling is expected to further deteriorate the overall scaling performance. From an HPC perspective, what is considered an overhead may vary. Arguably, inefficiencies due to load imbalance can be viewed as an overhead, e.g., attempting to use a resource set which provides high parallel efficiency for both solvers would likely result in a longer RE runtime. Whether this is an overhead or an inefficiency is a gray area. In any case, this can of course be alleviated by having different sets of resources for each solver (which is not explored in this paper). This in turn links to an undeniable overhead: the exchange of information between the solvers. When both solvers are mapped to the same hardware (i.e., with the same horizontal domain decomposition) as in these tests, this overhead is negligible. However, if states and fluxes must be exchanged through MPI across different resources (CPUs or GPUs), this overhead will grow.

[Figure]

**Figure 20.** (a) simulation time and (b) speedup for the Case 4 (3D domain). The black dashed line represents the ideal scaling

3. Rainfall is always applied to the SWE, even when no ponding exists. Since this is a modeling choice and may not hold in all situations, it would be good to clarify the limitations of this assumption. The Case 4 comparison shows the approach works well there, but a sentence noting scenarios where this may be less appropriate would be helpful.

[Response: Thanks for your suggestion. We have addressed the limitations in the revised manuscript with the following clarification in Section 3.4:]

Section 3.4 : It is important to note that in the SERGHEI-SWE-RE framework, rainfall is applied to the surface flow module regardless of whether surface ponding is generated. When asynchronous coupling is employed, this implementation may introduce discrepancies, as the surface–subsurface exchange flux lags behind the evolution of surface flow (Fig. 3). Specifically, rainfall may trigger surface runoff before the subsurface module has the opportunity to compute infiltration. In such circumstances, the use of asynchronous coupling is not recommended.

**Specific and Minor Comments:**

1. Line 125, "that" to "than"

[Response: Fixed.]

2. Figure 3 and the accompanying explanation are hard to follow. In the figure, if the intervals are meant to represent the time steps $\Delta t$, their lengths should be consistent. It is also not clear what the shaded/boxed "tsub" period represents. Does this interval mark the window during which the SWE and RE solvers exchange information? If so, this should be stated more explicitly in both the figure and the text.

[Response: We apologize for the confusion. We have redrawn Figure 3 and rewritten the corresponding context in Section 2.2 as follows:]

Section 2.2 :SERGHEI-SWE-RE supports both synchronous and asynchronous coupling between surface and subsurface modules. In synchronous coupling, the surface and subsurface modules are executed at every global time step, defined as the smaller of their individual time steps, ensuring both solvers advance on a common timeline. In asynchronous coupling, as shown in Fig. 3, the two modules operate on their own independent timelines, denoted as $T_{swe}^p = \sum_{i=0}^p \Delta t_{swe}^i$ and $T_{re}^q = \sum_{i=0}^q \Delta t_{re}^i$,

where $p$ and $q$ are the time step indices and $\Delta t^p_{swe}$ and $\Delta t^q_{re}$ are the corresponding time step sizes. Both $\Delta t^p_{swe}$ and $\Delta t^q_{re}$ vary in time. The former is restricted by the CFL condition, and the latter is adjusted based on either the maximum change in water content (in the PC scheme) or the number of linearization iterations (in the MP scheme). A user-defined upper bound, $\Delta t_{re,max}$, is enforced to further constrain $\Delta t^q_{re}$ (Li et al., 2025). The subsurface module is executed only when $T_{re} + \Delta t^q_{re} < T_{swe}$. With this approach, the SSE flux ($q_{ss}$) is computed using the current surface flow state and the previous subsurface flow state, shown as the red arrows in Fig. 3. Similarly, the latest surface water depth is sent backward to the subsurface module as boundary conditions, shown as the blue arrows in Fig. 3. Asynchronous coupling significantly enhances computational efficiency by reducing the execution times of the RE module. However, it introduces a time lag of the subsurface module as illustrated in Fig. 3. This time lag inevitably results in coupling errors that depend on both the time scale difference between surface and subsurface flow and the lag duration, an issue extensively discussed in (Li et al., 2023) and later in Section 3.5. Practically, users can tune $\Delta t_{re,max}$ to balance coupling accuracy and computational efficiency. It is, in principle, also possible to set the $\Delta t^q_{re} = N \Delta t^p_{swe}$ to prevent an excessive difference between the two time-step sizes. However, investigation of this approach, and a full investigation of the implications of synchronocity/asynchronicity is left for future work. Finally, it is worth noting that the synchronous/asynchronous coupling strategies apply to both PC and MP schemes of the RE solver. For both schemes, the surface water depth is used as a boundary condition for implicitly solving the subsurface pressure head. Consequently, the data exchange pathways between modules, shown as red and blue arrows in Fig. 3, are identical for both solution schemes.

[Figure]

**Figure 3.** Schematic of the asynchronous coupling strategy between surface and subsurface modules over time. The red arrows denote the transfer of the SSE flux ($q_{ss}$) as a volumetric source/sink term to the surface domain. The blue arrows denote the transfer of the surface water depth as the boundary condition to the subsurface domain.

3. In Section 2.3, the description of domain decomposition suggests that the model uses a structured grid without regional refinement. If this is indeed the case, it would be helpful to state this explicitly. Clarifying this will make it easier for readers to understand the limitations in the current domain decomposition strategy that you discuss later in the manuscript.

[Response: Thank you for your suggestion. We have added the following text to illustrate the grid system in the revised manuscript: ]

The model employs uniform structured grids in the horizontal directions for both surface and subsurface domains. Identical horizontal grid resolutions are used across both domains to allow the one-to-one coupling, as described in Section 2.2. Variable grid spacing is adopted in the vertical direction of the subsurface domain.

4. Line 174-176. The author should either report the results even in the supplement, or they should not use such a statement to support their conclusion.

[Response: We apologize for the omission of supporting results in the original manuscript. We have added a new figure to demonstrate the C-property.]

Fig. 5 illustrates that both the free surface elevation and subsurface pressure fields maintain their initial states during the entire simulation as expected, indicating that the C-property is satisfied.

[Figure]

**Figure 5.** The simulated free surface and groundwater table elevations for Case 1 at (a) T=0 s, (b) T=500 s, (c) T=1000 s on the transect of $y = 90$ m.

5. Line 179, what is the range of y for the inlet boundary?

[Response: We have added the following clarification in the revised manuscript:]

This test case maintains the same domain configuration as Case 1, except that a fluctuating surface water level is enforced at the inlet boundary located at $x$=260 m for $y$ ranges from 0 to 260 m, following the hydrograph shown in Fig. 6.

6. Line 246 "Fig. 14(b)" -> "Fig. 15(b)"

[Response: Fixed]

7. Lines 247–248: The explanation provided is reasonable, but I suggest elaborating a bit more here. Since simplified governing equations such as the kinematic and diffusive wave formulations are widely used in rainfall–runoff modeling, the distinction you highlight is important. This result that full SWE produces systematically larger ponding depths compared to the simplified approaches deserves to be emphasized more clearly, as it is likely to be of broad interest to the hydrologic modeling community.

[Response: We agree that this is indeed a very relevant point. We have thus expanded on the discussion (Section 3.5), which is contextualised with previous results reported in the literature which are consistent with our observations and support our conclusions. We also point out that further detailed investigations of these differences introduced by the different surface modelling approaches are warranted, both strictly in the surface flow modelling realm as in the coupled surface-subsurface context. We have also brought this issue in our conclusions.]

Section 3.5: This indicates that in this case, the simplified equations underestimate the water depth compared to the full SWE, thereby causing less ponding storage. Consistent with this, Caviedes-Voullième et al. (2020) and Li and Hodges (2021) reported that the diffusive wave approximation systematically underestimates water depths compared to the full SWE model in rainfall-runoff simulations. Similarly, de Almeida and Bates (2013) showed that simplified models (including diffusive wave equations and local inertial models) underestimate water depth gradients under unsteady conditions, leading to reduced ponding. Therefore, the model discrepancy in Fig. 15(b) does not undermine the validation of SERGHEI-SWE-RE for reliable surface–subsurface exchange simulations. In contrast, it underscores the need for the full SWE to resolve surface water dynamics, which is essential for accurate surface–subsurface exchange modeling. It also prompts the need to further conduct cross-model comparisons for coupled surface-subsurface models under more dynamic flow conditions, since kinematic and diffusive wave approaches are far more prevalent.

Section 5: The results also suggest potential differences between surface-subsurface models depending on how surface flow is formulated, i.e., typically kinematic and diffusive wave approaches in contrast to the fully dynamic approach adopted herein.

This warrants further investigation.

8. For Figures 15 and 19, several curves overlap almost exactly, which makes them hard to distinguish. In Fig. 15, the blue and red lines lie directly on top of each other; using different line thicknesses (or slightly different styles) would make the overlap clearer. Similarly, in Fig. 19, the black circular markers are difficult to see because they coincide with another line. Adjusting line weights or marker sizes would improve readability in both figures.

[Response: We apologize for the insufficient visual distinction in the figures and thank you for your helpful suggestions regarding the plot presentation. We have revised Figures 15 and 19 in the revised manuscript accordingly.]

[Figure]

**Figure 15.** The temporal variations in (a) outlet discharge and (b) surface water volume. SERGHEI represents the synchronous calculation; SERGHEI-asy-4s represents the asynchronous calculation with the $\Delta t_{re,max} = 4s$. SERGHEI-slab2 is a customized SWE-only simulation on the slab 2, with the upslope and downslope sections truncated.

[Figure]

**Figure 19.** Parallel efficiency for the weak scaling tests performed with the extended lateral flow problem, on the JUWELS Booster with Nvidia A100 GPUs. Full, SWE and RE in the legends represent the efficiency of the entire simulation, the SWE solver and the RE solver, respectively. The dashed line represent the ideal efficiency.